# Computationally defined and in vitro validated putative genomic safe harbour loci for transgene expression in human cells

Matias I Autio[1,2,3]\*, Efthymios Motakis[2†‡], Arnaud Perrin[1,2†], Talal Bin Amin[3], Zenia Tiang[1,2], Dang Vinh Do[1,2], Jiaxu Wang[4], Joanna Tan[5], Shirley Suet Lee Ding[6], Wei Xuan Tan[6,7], Chang Jie Mick Lee[1,2], Adrian Kee Keong Teo[6,7,8], Roger SY Foo[1,2]\*

[1]Laboratory of Molecular Epigenomics and Chromatin Organization, Genome Institute of Singapore, Singapore, Singapore; [2]Cardiovascular Disease Translational Research Programme, Yong Loo Lin School of Medicine, Singapore, Singapore; [3]Laboratory of Systems Biology and Data Analytics, Genome Institute of Singapore, Singapore, Singapore; [4]Laboratory of RNA Genomics and Structure, Genome Institute of Singapore, Singapore, Singapore; [5]Center for Genome Diagnostics, Genome Institute of Singapore, Singapore, Singapore; [6]Stem Cells and Diabetes Laboratory, Institute of Molecular and Cell Biology, Singapore, Singapore; [7]Department of Medicine, Yong Loo Lin School of Medicine, National University of Singapore, Singapore, Singapore; [8]Precision Medicine Translational Research Programme, Department of Biochemistry, Yong Loo Lin School of Medicine, National University of Singapore, Singapore, Singapore

\*For correspondence:
autiomi@gis.a-star.edu.sg (MIA);
roger.foo@nus.edu.sg (RSYF)

†These authors contributed equally to this work

Present address: ‡The Jackson Laboratory for Genomic Medicine, Computational Sciences Farmington, CT, United Kingdom

**Abstract** Selection of the target site is an inherent question for any project aiming for directed transgene integration. Genomic safe harbour (GSH) loci have been proposed as safe sites in the human genome for transgene integration. Although several sites have been characterised for transgene integration in the literature, most of these do not meet criteria set out for a GSH and the limited set that do have not been characterised extensively. Here, we conducted a computational analysis using publicly available data to identify 25 unique putative GSH loci that reside in active chromosomal compartments. We validated stable transgene expression and minimal disruption of the native transcriptome in three GSH sites in vitro using human embryonic stem cells (hESCs) and their differentiated progeny. Furthermore, for easy targeted transgene expression, we have engineered constitutive landing pad expression constructs into the three validated GSH in hESCs.

## Editor's evaluation

This study presents solid data on the computational identification of 25 putative human genomic safe harbor loci, of which 3 have been experimentally validated using human embryonic stem cells, that may serve as safe sites for transgene integration. These findings will be an invaluable resource in cell and gene therapy applications. This work will be of interest to cell biologists and researchers in the stem cell field.

## Introduction

Stable expression of transgenes is essential in both therapeutic and research applications. Traditionally, transgene integration has been accomplished via viral vectors in a semi-random fashion, but with inherent integration site biases linked to the type of virus used (*Mitchell et al., 2004*). The randomly integrated transgenes may undergo silencing (*Ellis, 2005*; *Mok et al., 2007*) and more concerningly, can also lead to dysregulation of endogenous genes. Gene dysregulation can lead to malignant transformation of cells and has unfortunately given rise to cases of leukaemia (*Howe et al., 2008*; *Hacein-Bey-Abina et al., 2003*) in gene therapy trials. GSH loci have been previously suggested as safe sites for transgene integration. Criteria proposed for a putative GSH include; a set distance from coding and non-coding genes; with added separation from known oncogenes and miRNAs, and no disruption of transcriptional units or ultra conserved regions (*Papapetrou and Schambach, 2016*; *Papapetrou et al., 2011*; *Sadelain et al., 2011*). To date, a number of sites in the human genome have been used for directed integration; however none of these pass scrutiny as bona fide GSH (*Papapetrou and Schambach, 2016*). Here, we conducted a computational analysis to filter sites that meet criteria for GSH loci. In addition to the safety criteria, we identified regions that reside in active chromosomal compartments in many human cell and tissue types. Our analysis yielded a final list of 25 unique putative GSH that are predicted to be accessible in multiple cell types. We used human embryonic stem cells (hESCs) and their differentiated progeny to validate stable transgene expression in three of the putative GSH sites in vitro. Furthermore, to enable easy targeted transgene expression, we generated hESC lines with constitutive landing pad expression constructs targeted into the three validated GSH.

## Results

### Computational filtering for safe and accessible loci

A list of criteria has previously been suggested for a given locus to qualify as putative GSH (*Papapetrou and Schambach, 2016*; *Sadelain et al., 2011*). These criteria state that GSH: must not be in proximity to genes coding or non-coding, with added distance from known oncogenes and miRNAs, and must not disrupt transcriptional units or ultra-conserved genomic regions. To shortlist putative GSH we conducted a computational search of the human genome using publicly available data (*Figure 1A*). We included the previously published safety criteria and added a further filter to exclude any regions of DNaseI hypersensitivity, as these regions are likely enriched in transcription factor binding and regulatory elements (*Meuleman et al., 2020*). A total of 12,766 sites, ranging from 1 b to approximately 30 Mb, passed the filters used (*Figure 1A–B*; *Supplementary file 1*). For a universal GSH site to be useful, it needs not only to be safe, but also enable stable expression of a transgene in any tissue type. We filtered the human genome for regions consistently in the active chromatin compartment based on 21 different human cell and tissue types (*Schmitt et al., 2016*). To extend the analysis beyond the limited set of samples, we utilised RNA-seq data of all available tissue types from the GTEx portal (*Carithers et al., 2015*). We selected an empirical set of ubiquitously expressed genes with low variance. We then cross referenced the chromosomal locations of these genes with the consistently active chromatin regions. This analysis yielded 399 1 Mb active regions that overlapped with a ubiquitously expressed gene (*Figure 1A–B*; *Supplementary file 1*). By overlapping the two datasets, we found 49 safe sites within the active regions. We further filtered the 49 sites (*Figure 1A–B*; *Supplementary file 1*) using BLAT to remove candidate sites that had highly similar sequence matches at other genomic loci and generated a final shortlist of 25 unique putative universal GSH sites in the human genome (*Figure 1A–B*; *Table 1*).

### Targeted knock-in at putative GSH with CRISPR/Cas9

To validate our candidate GSH, we selected 7 of the 25 sites at random for in vitro experiments (*Figure 1C*). None of the selected seven sites lie at or immediately adjacent to borders of topologically associated domains (TADs; *Figure 1—figure supplements 1–7*). We targeted H1 hESC using CRISPR/Cas9 and a donor landing pad construct (*Figure 2*) at each of the seven candidate sites (Methods and *Supplementary file 2*). To minimise potential off-target effects of CRISPR/Cas9 targeting, we used a version of Cas9 with enhanced specificity (*Slaymaker et al., 2016*) and effective guides with highest predicted specificity available. Following antibiotic selection, single clones were expanded and screened for successful homology directed repair driven integration of the expression construct with

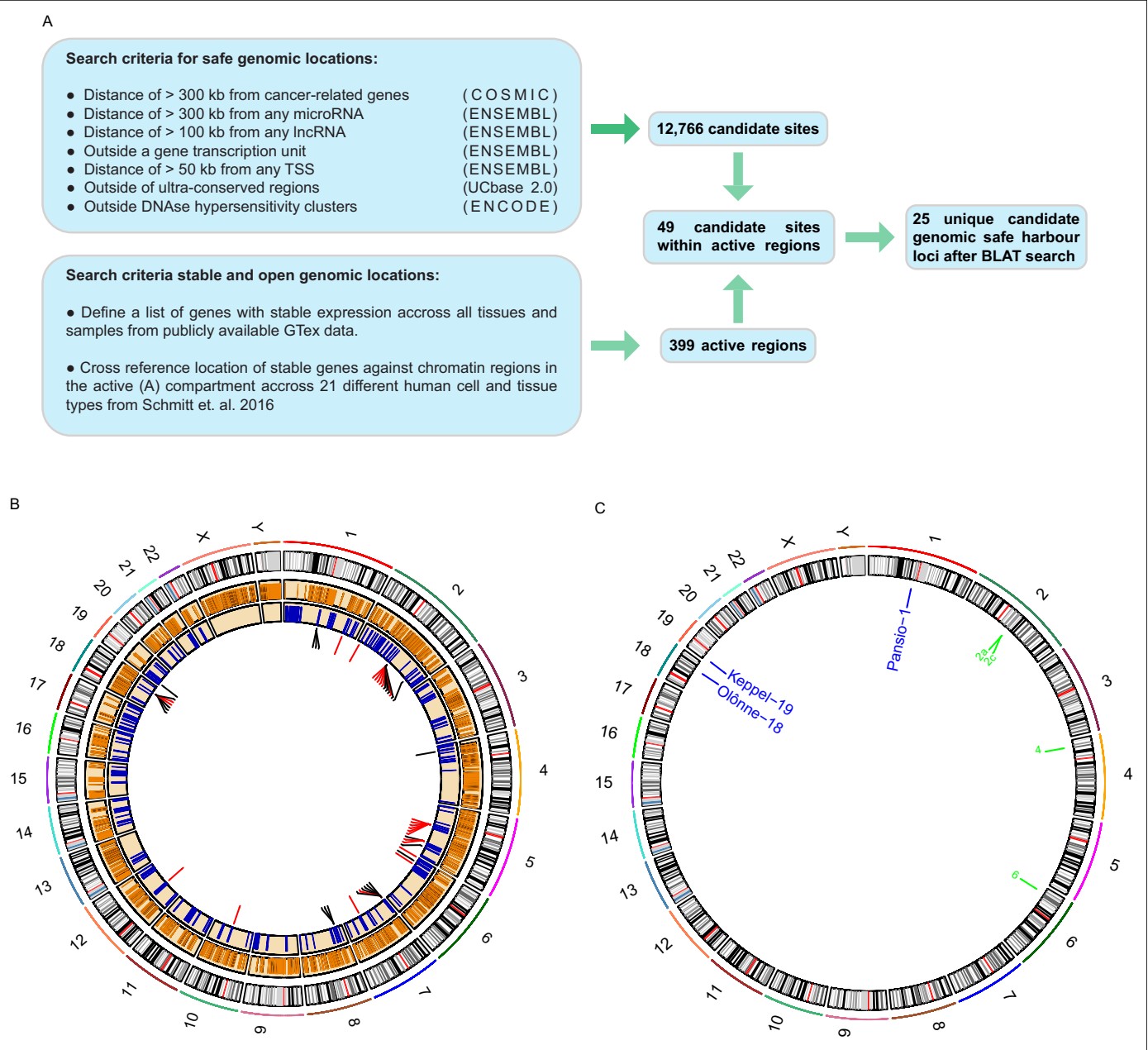

**Figure 1.** Computational search for candidate GSH. (**A**) Schematic representation of the computational workflow for defining candidate GSH. (**B**) CIRCOS plot summarising computational search results. Ring 1: chromosome ideograms; ring 2: orange bars indicating safe sites; ring 3: blue bars indicating active regions; ring 4: candidate sites within active regions, red bars site failed BLAT screening, black bars site passed BLAT screening. (**C**) Locations of candidate GSH targeted in vitro. Blue labels: targeted clone established; green labels: no clone established.

The online version of this article includes the following figure supplement(s) for figure 1:

**Figure supplement 1.** Hi-C profile for GSH candidate 1.

**Figure supplement 2.** Hi-C profile for GSH candidate 2a.

**Figure supplement 3.** Hi-C profile for GSH candidate 2c.

**Figure supplement 4.** Hi-C profile for GSH candidate 4.

**Figure supplement 5.** Hi-C profile for GSH candidate 6.

**Figure supplement 6.** Hi-C profile for GSH candidate 18.

**Figure supplement 7.** Hi-C profile for GSH candidate 19.

**Table 1.** Coordinates of candidate GSH, their associated active chromosome regions & housekeeping gene, and BLAT score against the most similar region.

| Chromosome | Start | End | Width | Active region start | Active region end | Housekeeping gene | BLAT-score |
|---|---|---|---|---|---|---|---|
| 1 | 113289036 | 113289342 | 307 | 113000001 | 114000000 | HIPK1 | 0.21 |
| 1 | 113314841 | 113318369 | 3529 | 113000001 | 114000000 | HIPK1 | 0.18 |
| 1 * | 113339961 | 113340514 | 554 | 113000001 | 114000000 | HIPK1 | 0.27 |
| 2 * | 128912721 | 128914814 | 2094 | 128000001 | 129000000 | UGGT1 | 0.08 |
| 2 | 128918961 | 128919839 | 879 | 128000001 | 129000000 | UGGT1 | 0.16 |
| 2* | 128932307 | 128935799 | 3493 | 128000001 | 129000000 | UGGT1 | 0.15 |
| 2 | 128963272 | 128965759 | 2488 | 128000001 | 129000000 | UGGT1 | 0.44 |
| 2 | 208992998 | 208997459 | 4462 | 208000001 | 209000000 | PIKFYVE | 0.28 |
| 4* | 17373361 | 17374159 | 799 | 17000001 | 18000000 | MED28 | 0.04 |
| 5 | 131058585 | 131058947 | 363 | 131000001 | 132000000 | FNIP1 | 0.41 |
| 5 | 148753741 | 148757219 | 3479 | 148000001 | 149000000 | FBXO38 | 0.10 |
| 6 * | 15727241 | 15727490 | 250 | 15000001 | 16000000 | DTNBP1 | 0.47 |
| 7 | 4314741 | 4315279 | 539 | 4000001 | 5000000 | FOXK1 | 0.15 |
| 7 | 4321017 | 4323839 | 2823 | 4000001 | 5000000 | FOXK1 | 0.09 |
| 7 | 4328040 | 4329659 | 1620 | 4000001 | 5000000 | FOXK1 | 0.21 |
| 7 | 4353504 | 4354219 | 716 | 4000001 | 5000000 | FOXK1 | 0.32 |
| 7 | 4454808 | 4456201 | 1394 | 4000001 | 5000000 | FOXK1 | 0.17 |
| 8 | 23945241 | 23945819 | 579 | 23000001 | 24000000 | R3HCC1 | 0.34 |
| 8 | 23986981 | 23988319 | 1339 | 23000001 | 24000000 | R3HCC1 | 0.06 |
| 8 | 23999628 | 24001194 | 1567 | 23000001 | 24000000 | R3HCC1 | 0.02 |
| 18 | 56339813 | 56340245 | 433 | 56000001 | 57000000 | TXNL1 | 0.06 |
| 18 | 56396821 | 56397319 | 499 | 56000001 | 57000000 | TXNL1 | 0.07 |
| 18 | 56410681 | 56411039 | 359 | 56000001 | 57000000 | TXNL1 | 0.14 |
| 18 * | 56534775 | 56536439 | 1665 | 56000001 | 57000000 | TXNL1 | 0.16 |
| 19 * | 5400761 | 5402139 | 1379 | 5000001 | 6000000 | SAFB | 0.18 |

*= GSH shortlisted for in vitro validation.

junction- and digital-PCR (*Figure 2B*, *Supplementary file 3*, and *Figure 2—figure supplement 1*). Successful heterozygous targeting of the donor construct was confirmed at three candidate GSH sites on chromosomes 1, 18, and 19. These three GSH sites were subsequently also targeted in H9 hESC for validation in an independent cell line (*Figure 2—figure supplement 1*, *Supplementary file 3*). No evidence of off-target activity was observed following PCR amplification and Sanger sequencing of the top five predicted off-target sites for each of the targeted clones (*Figure 2—figure supplements 2–4*). We named the successfully targeted safe harbours after real world harbours, designating them Pansio-1, Olônne-18, and Keppel-19.

## In vitro validation of targeted GSH in hESCs

To investigate the safety of our targeted GSH, we first checked the mRNA expression levels of the nearest genes *MAGI3*, *TXNL1* and *ZNRF4* to Pansio-1, Olônne-18 and Keppel-19 respectively using qPCR. When compared to un-targeted H1 hESCs, the mean log2 fold-change (log2-FC) values of the three nearest genes in the three H1 GSH clones as well as an independent un-targeted H1 sample ranged from –0.067 to 0.065 (*Figure 2C*, *Supplementary file 4*). The results from same comparison

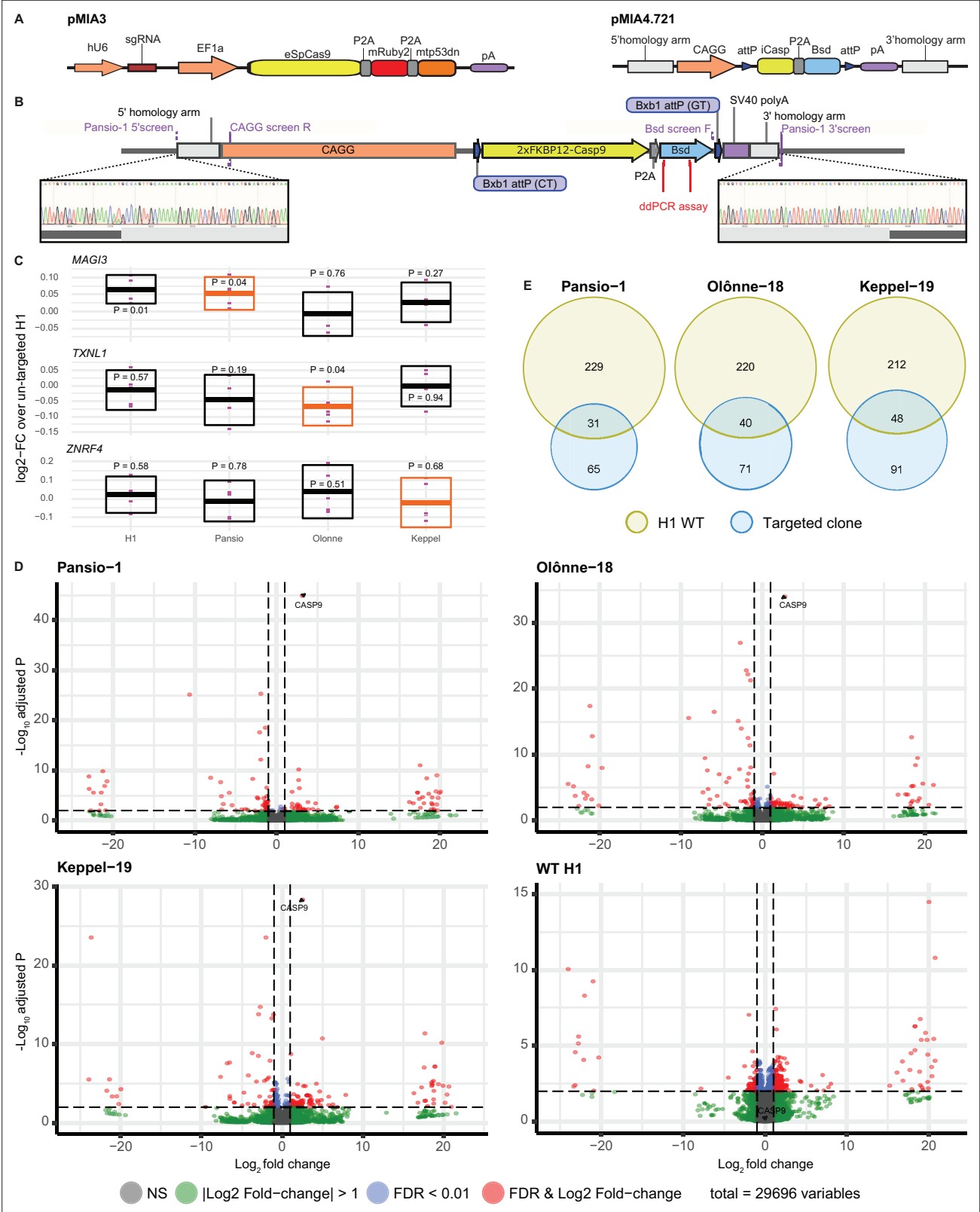

**Figure 2.** GSH targeting and transcriptomic analysis of H1 hESC. (**A**) Schematic representation of CRISPR/Cas9 plasmid (pMIA3) and homology directed repair donor (pMIA4.721) used for targeting with functional components annotated. (**B**) Schematic of integrated landing pad expression construct. Positions of primers for junction-PCR as well as of ddPCR assay are indicated. Representative junction-PCR Sanger sequencing reads from Pansio-1 targeted clones shown in expanded view. (**C**) Log2-FC of mRNA expression levels against un-targeted H1 hESC samples for the nearest genes of

*Figure 2 continued on next page*

*Figure 2 continued*

Pansio-1, Olônne-18, and Keppel-19 candidate GSH. Evaluated samples: H1=un-targeted hESC, Pansio-1=landing pad construct integrated to Pansio-1 GSH in H1 hESC, Olônne-18=landing pad construct integrated to Olônne-18 GSH in H1 hESC and Keppel-19=landing pad construct integrated to Keppel-19 GSH in H1 hESC. Box plots representing 95% confidence intervals of mean log2-FC. Nearest gene for each GSH indicated in orange. Individual data points shown in pink with p-value for each comparison shown above. (**D**) Volcano plots of RNA-seq analysis against un-targeted H1 hESC. Samples analysed as in (**C**). Differentially expressed (DE) genes with FDR ≤0.01 and |logFC|≥1 in pink, genes with |logFC|≥1 in green, genes with FDR ≤0.01 in blue, others in grey. (**E**) Venn-diagrams illustrating the overlap of DE genes between un-targeted H1 hESC and the three GSH targeted H1 hESC lines.

The online version of this article includes the following figure supplement(s) for figure 2:

**Figure supplement 1.** PCR gel images of junction PCR and wild type allele PCR reactions for screened clones.

**Figure supplement 2.** Off-target screening for Pansio-1.

**Figure supplement 3.** Off-target screening for Olônne-18.

**Figure supplement 4.** Off-target screening for Keppel-19.

**Figure supplement 5.** Transcriptomic analysis of GSH targeted H9 hESC.

**Figure supplement 6.** Representative images of metaphase spreads used for karyotyping the GSH targeted H1 and H9 cell lines.

**Figure supplement 7.** Representative images of haematoxylin and eosin stained teratoma from Pansio-1, Olônne-18, and Keppel-19 H1 cells.

**Figure supplement 8.** Inducible expression from GSH integrated cassette.

**Figure supplement 9.** Schematic representation of the pMIA10.53-Clover donor plasmid.

with our established H9 lines ranged from log2-FC of –0.14 to 0.04 (*Figure 2—figure supplement 5*, *Supplementary file 4*). Only three one sample Student's t-tests on the log2-FC results showed p-values of less than 0.05, including the H1 vs. H1 comparison of *MAGI3* (*Figure 2C*, *Figure 2—figure supplement 5*, *Supplementary file 4*). Overall, the nearest genes to our targeted GSH sites showed minimal change in expression levels when compared to untargeted cells.

We then conducted RNA-seq analysis to look for gene expression changes on a global scale. Pairwise comparisons of our GSH targeted clones to un-targeted H1 hESCs yielded very low numbers of differentially expressed (DE) genes; 96, 111, and 119 out of 29,696 observed genes respectively for Pansio-1, Olônne-18, and Keppel-19 and 260 genes for independent H1 un-targeted samples (*Figure 2D* and *Supplementary file 5*). Notably *CASP9*, the suicide gene included in our targeting construct, was the gene with lowest false discovery rate (FDR) in all three GSH lines (*Figure 2D* and *Supplementary file 5*). A high proportion of the DE genes found in our GSH targeted lines were shared with the DE genes found from pair-wise comparisons of the independent set of wild-type H1 cells to the un-targeted H1 hESCs (*Figure 2E* and *Supplementary file 5*). Same RNAseq analysis on the H9 GSH targeted lines also yielded a low number of DE genes; 60, 51, and 91 out of 35,070 observed genes respectively for Pansio-1, Olônne-18, and Keppel-19 (*Figure 2—figure supplement 5* and *Supplementary file 6*). Pairwise comparison of the independent un-targeted H9 samples to the control H9 samples yielded 15 DE genes (*Supplementary file 6*). *CASP9* was identified as a DE gene in all GSH samples, although unlike in the H1 clones in H9 it did not have the lowest FDR in any comparison (*Supplementary file 6*). We also compared the genomic locations of all the observed DE genes from our RNAseq analysis to the GSH targeting locations. The closest DE gene found from our analysis in both H1 and H9 cells lies >3 Mb away for Pansio-1, >5 Mb for Olônne-18 and approximately 1 Mb for Keppel-19 (*Supplementary files 5 and 6*). None of the DE genes shared across the H1 and H9 cells are located on the same chromosome as the respective GSH loci that were targeted (*Supplementary file 5*), except *CASP9*, part of our targeting construct, and Pansio-1 on chromosome 1.

Functional enrichment analysis of the DE genes revealed relatively few terms; 2, 15, and 25 in H1 cells and 6, 5, and 18 in H9 for Pansio-1, Olônne-18, and Keppel-19, respectively (*Supplementary files 5 and 6*). Many of the terms highlighted by the functional enrichment included the *CASP9* gene, which is over expressed from our targeting construct (*Supplementary files 5 and 6*). As a further safety check, we also conducted karyotyping of the established H1 and H9 lines and observed no abnormalities (*Figure 2—figure supplement 6*). We also used the Pansio-1, Olônne-18, and Keppel-19 H1 hES lines to generate teratomas in immunodeficient mice. All three lines were able to give rise to tissues from endoderm, ectoderm, and mesoderm lineages (*Figure 2—figure supplement 7*). Taken

together, these results suggest minimal disruption of the native genome or the pluripotent cell identity and show a lack of karyotypic abnormalities following transgene integration to our three GSH.

In addition to safety, a functional GSH also needs to allow for stable expression of a transgene. We took advantage of the landing-pad design of our targeting construct to swap in a sequence coding for Clover-fluorophore, by introducing a plasmid expressing BxbI-integrase as well as a donor construct into the three GSH lines (*Figure 3A & B*) in both H1 and H9 cells. Targeted cells were enriched with fluorescence activated cell sorting. Introduction of the payload transgene did not alter the expression of pluripotency markers OCT3/4 and SOX2 (*Figure 3C–D*, *Figure 3—figure supplement 1A–B*). We maintained the Clover targeted GSH lines in hESC state over 15 passages and consistently observed >98% Clover-positive cells (*Figure 3E*, *Figure 3—figure supplement 1C*). To investigate the stability of our GSH in other cell types, we conducted directed differentiation of our Clover-integrated hESC lines into cell types from the three germ lineages. Clover expression remained consistent in neuronal, liver, and cardiac cells (*Figure 3F–H*, *Figure 3—figure supplements 1D and 2*) in Pansio-1, Olônne-18, and Keppel-19 targeted H1 and H9 cells. In addition to neuronal, liver, and cardiac lineages, we also differentiated cells to pancreatic β-cells using the Pansio-1 targeted H9 line and observed Clover-transgene expression in marker-positive cells (*Figure 3—figure supplement 1E*). To further quantify the Clover-transgene expression in differentiated cell types, we conducted high-content imaging on the Pansio-1 Olônne-18, and Keppel-19 targeted H9 cells differentiated to neuronal, liver, and cardiac lineages (*Figure 4*). Quantification of the captured images indicated high correlation of the lineage markers and the Clover transgene signal, ranging from 75% to 99% (*Figure 4—figure supplement 1* and *Supplementary file 7*). The expression from our landing-pad cassette is driven by a constitutive CAGG promoter (*Figure 3A & B*). To study whether our candidate GSH loci support transgene expression driven by other promoter sequences, we constructed an 'all-in-one' donor cassette for Tet-inducible transgene expression (Methods, *Figure 2—figure supplement 8A*). H1 Pansio-1 hESC successfully targeted with the donor cassette were first selected with puromycin and then treated with doxycycline. Twenty-four hour incubation with 1 μg/ml doxycycline was sufficient to drive expression of the GFP transgene on average in 73% of cells (*Figure 2—figure supplement 8C–E*, *Supplementary file 8*). Results from the above experiments indicate that our landing-pad GSH cell lines are able to support stable transgene expression under the CAGG promoter in hESC and derived cell types and that inducible expression from a tet-driven promoter can be observed in hESC.

Overall, we have developed a computational pipeline to define GSH candidate sites from the human genome that fulfil criteria for safety as well as accessibility for transgene expression. Our pipeline defines 25 unique candidate GSH and we conducted in vitro validation experiments for three of them, Pansio-1, Olônne-18, and Keppel-19. Targeting and transgene expression in hESC at the three sites led to minimal or no change in the expression levels of the nearest native genes or the transcriptome overall and did not interfere with directed differentiation to the three germ lineages. The three tested GSH support transgene expression in the pluripotent state and in derived cell types from all the germ lineages. Furthermore, we established landing pad expression lines in H1 and H9 hESC of Pansio-1, Olônne-18, and Keppel-19, which we hope will serve as useful research tools.

## Discussion

A GSH site is an ideal location for transgene integration. To qualify as a GSH, a locus should be able to host transgenes enabling their stable expression as well as not interfere with the native genome (*Papapetrou and Schambach, 2016*). A number of previous studies have reported discovery and usage of a handful of integration sites (*Papapetrou et al., 2011*; *Costa et al., 2005*; *Pellenz et al., 2019*; *Rodriguez-Fornes et al., 2020*; *Eyquem et al., 2013*), which fulfil a subset of criteria previously suggested for GSH (*Papapetrou and Schambach, 2016*; *Sadelain et al., 2011*). More recent publications have utilised a complete set of the suggested GSH criteria to identify integration sites (*Aznauryan et al., 2022*; *Odak et al., 2023*). However, in contrast to the candidate sites we present (*Figure 1A–B*), the previously reported sites do not utilise criteria to avoid potential regulatory elements or criteria for universally stable and active genomic regions (*Papapetrou et al., 2011*; *Costa et al., 2005*; *Pellenz et al., 2019*; *Rodriguez-Fornes et al., 2020*; *Eyquem et al., 2013*; *Aznauryan et al., 2022*; *Odak et al., 2023*). We utilised directed differentiation of our Pansio-1, Olônne-18 and Keppel-19 targeted hESC to show consistent expression in hESC and cells from all three germ

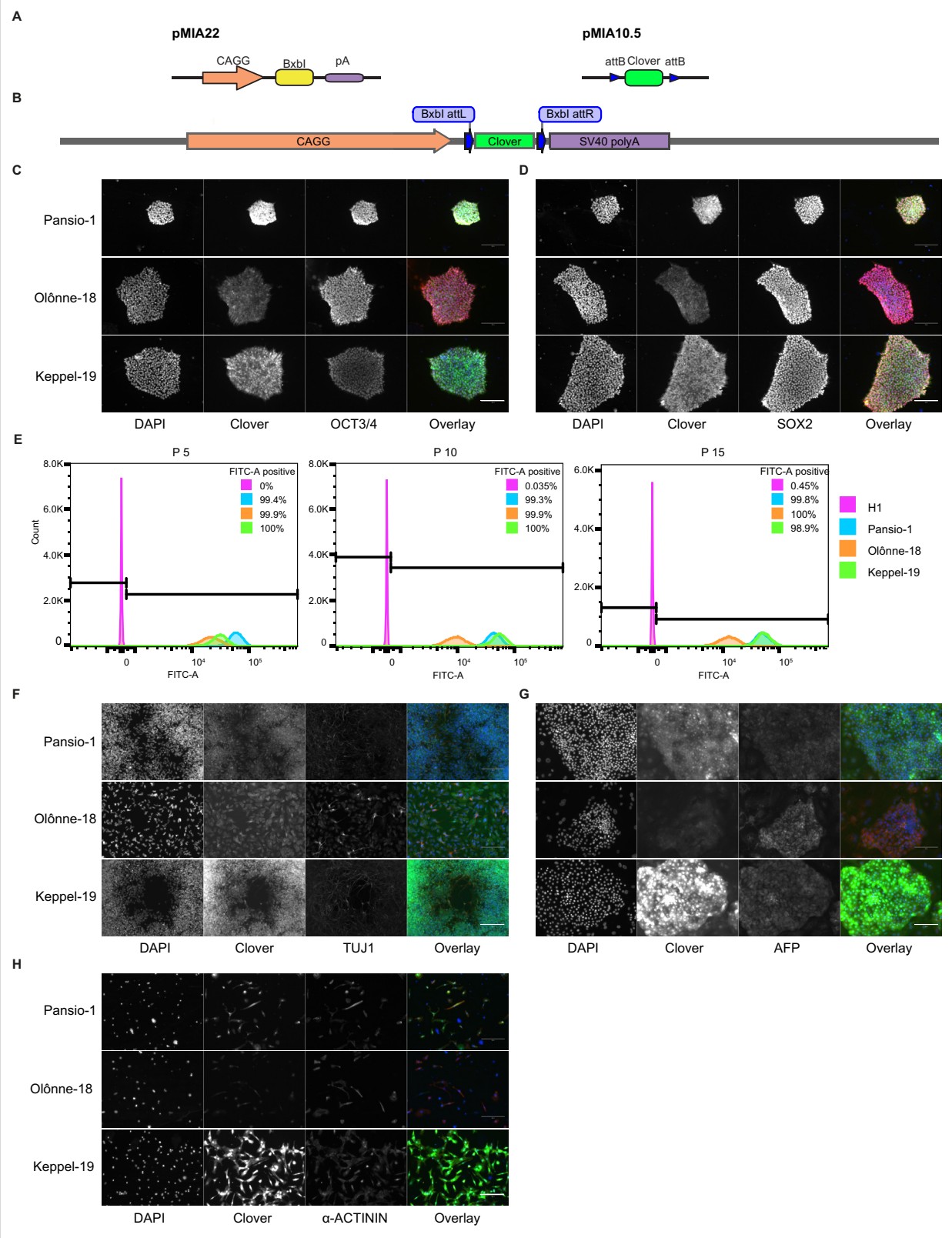

**Figure 3.** Integration and validation of transgene expression in GSH targeted H1 hESC and their differentiated progeny. (**A**) Schematic representation of integrase expression construct (pMIA22) and transposon donor construct (pMIA10.5). (**B**) Schematic of landing pad construct with integrated Clover transgene. (**C**) Representative immunofluorescence images of Clover-integrated GSH H1 cells. DAPI = nuclear staining with 4',6-diamidino-2-phenylindole, Clover = fluorescence from Clover transgene, OCT3/4=antibody staining against OCT3/4, Overlay = overlay of the three imaged

*Figure 3 continued on next page*

*Figure 3 continued*

channels. (**D**) As in (**C**) apart from antibody staining against SOX2. (**E**) Histograms of flow cytometry analysis for FITC-A channel of un-targeted H1 hESC, and the three GSH targeted hESC lines over 15 passages. Percentages of FITC-A-positive cells according to the indicated gating. (**F**) Representative immunofluorescence images of Clover-integrated GSH H1 cells differentiated to neuronal-like cells. Channels imaged as in (**C**) apart from antibody staining against TUJ1. (**G**) As in (**F**) for cells differentiated to hepatocyte-like cells, antibody staining against AFP. (**H**) As in (**F**) for cells differentiated to cardiomyocyte-like cells, antibody staining against sarcomeric α-ACTININ. Scale bars for all immunofluorescence images equal to 150 μm.

The online version of this article includes the following figure supplement(s) for figure 3:

**Figure supplement 1.** Integration and validation of transgene expression in GSH targeted H9 hESC and their differentiated progeny.

**Figure supplement 2.** Representative images of H1 Pansio-1 line immunofluorescence staining with AF594 secondary antibody alone.

lineages (*Figure 3C–H*), whereas majority of the previously reported sites remain studied in only a limited number of cell types (*Pellenz et al., 2019*; *Rodriguez-Fornes et al., 2020*; *Eyquem et al., 2013*; *Aznauryan et al., 2022*; *Odak et al., 2023*).

The integration sites that have been most heavily utilised in research, namely AAVS1, CCR5, and Rosa26 orthologous site (*Irion et al., 2007*; *Liu et al., 1996*; *Kotin et al., 1992*; *Perez et al., 2008*), do not meet the criteria set out for GSH (*Papapetrou and Schambach, 2016*; *Sadelain et al., 2011*). These sites reside in highly gene rich regions and in the case of AAVS1 actually within a gene transcription unit, furthermore all of these loci have known oncogenes in their proximity (<300 kb) (*Sadelain et al., 2011*). Thus, their utilisation in a clinical setting would require extensive further safety data. Furthermore variable transgene expression and silencing has been reported for AAVS1 in hepatocytes (*Ordovás et al., 2015*) and cardiomyocytes (*Bhagwan et al., 2019*). The three candidate sites we tested, Pansio-1, Olônne-18, and Keppel-19, demonstrated stable expression of the Clover transgene in hES cells and from all three germ-lineages identified by two lineage-specific markers after differentiation from hESC, including both hepatocyte- and cardiomyocyte-like cells (*Figure 3C–H*, *Figure 4*, and *Figure 3—figure supplement 1*).

The transcriptome analysis of our GSH targeted hESC lines showed very low number of differentially expressed genes (*Figure 2D–E*, *Figure 2–figure supplement 5*, *Supplementary files 5 and 6*) when compared to untargeted H1 or H9 hESC. Furthermore, many of the DE genes we observed were shared with independent wild type hESC (*Figure 2E* and *Figure 2—figure supplement 5*). The nearest observed DE genes to each targeted candidate GSH were located >1 Mb away from the targeted site, which is beyond the distance generally suggested for enhancer-promoter interactions (*Jerkovic and Cavalli, 2021*). If the transgene integration to our GSH candidate site had a direct *cis*-regulatory effect on expression of nearby genes, it would be expected that the genes affected would be shared between both the H1 and H9 cells in each respective GSH. The number of DE genes shared between the clones from the two cell lines is very low; 6, 9, and 8 for Pansio-1, Olônne-18, and Keppel-19, respectively (*Supplementary file 5*) and *CASP9* is the only shared DE gene from all the GSH targeted lines that is located on the same chromosome as the respective GSH site. However, as *CASP9* is over expressed as part of our targeting construct, the effect seen is unlikely due to activation of the endogenous transcript. Overall, the transcriptomic data does not indicate direct *cis*-regulatory effect on the expression levels of any genes from the transgene integration into targeted candidate GSH sites Pansio-1, Olônne-18, and Keppel-19. A more likely plausible explanation to the differences observed in our qPCR and RNA sequencing data could be heterogeneity arising from the derivation of clonal lines in our targeting process.

We show that the three tested candidate GSH sites are able to support constitutive transgene expression both in the pluripotent state and in derived cell types from all the three germ lineages. Furthermore, our data indicates that Pansio-1, Olônne-18, and Keppel-19 sites are able to support inducible expression in the pluripotent state. Previous studies have indicated the complex interplay between the site of transgene integration and the components of the expression cassette (promoter, enhancer, and insulators; *Ordovás et al., 2015*; *Odak et al., 2023*). Further experiments using our established cell lines may help to shed light on this interaction in the future.

Our data suggest that the three candidate GSH, Pansio-1, Olônne-18, and Keppel-19, are able to support stable transgene expression in different cell types, and integration at these sites shows minimal perturbation of the native transcriptome. As hESC currently offer the best available model to test the stability of transgene expression from a GSH in the human genome, we generated landing-pad H1 and H9 hESC lines for all three candidate sites. These cell lines will allow easy integration

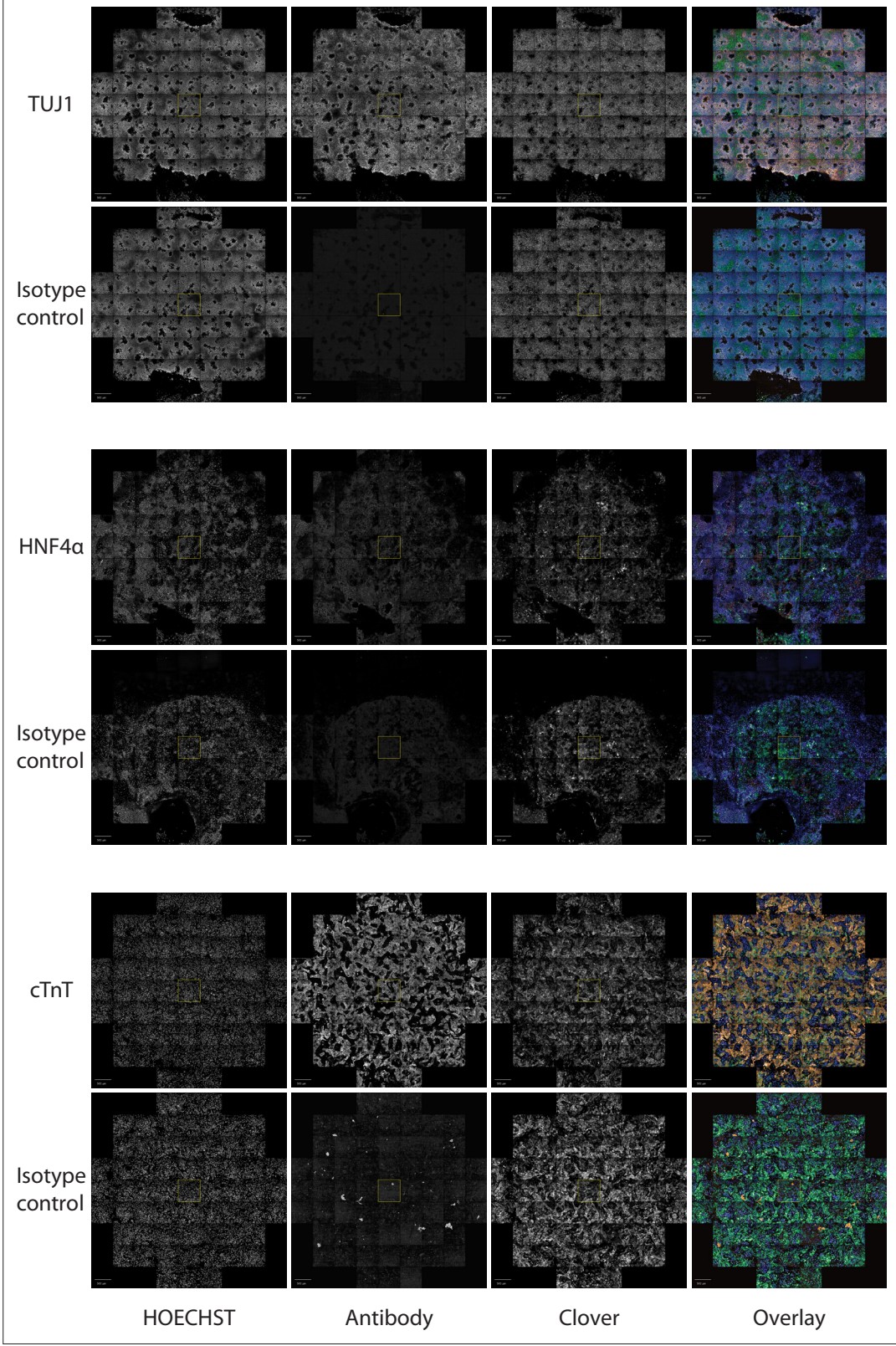

**Figure 4.** Representative images of Pansio-1 H9 cells differentiated to neuronal, hepatic, and cardiac cell types. Staining for respective lineage markers TUJ1, HNF4α and cTnT and isotype controls is shown as well as nuclear staining with HOECHST and the channel for Clover-transgene. Images are composites of 61 individual images from HCI. Scale bars in all images equal to 500 μm.

*Figure 4 continued on next page*

*Figure 4 continued*
The online version of this article includes the following figure supplement(s) for figure 4:
**Figure supplement 1.** Bar plots representing the mean ratio of positive cells from the high content imaging analysis.

of various transgenes to the candidate GSH for research applications. Advances in genome engineering technologies using for example e.g. novel integrases or CRISPR RNA-guided transposons (*Durrant et al., 2023*; *Yarnall et al., 2023*; *Lampe et al., 2023*; *Anzalone et al., 2022*) will likely lead to easier integration of more complex transgene constructs. We hope that our list of 25 candidate GSH and the three in vitro validated sites will serve as a resource for research applications in many different cell types. After further experimentation and reproducibility validations, translation to clinical applications can be envisaged.

## Methods
### Short-listing of putative safe harbour genomic regions
We applied a series of computational, whole-genome loci filtering criteria to pick a narrow list of high-confidence, putative safe harbour sites for experimental validation. In the first step, we selected genomic regions that satisfy simultaneously all of the following criteria: the loci should be located outside of ultra-conserved regions (*Lomonaco et al., 2014*; *Taccioli et al., 2009*) (coordinates lifted over from hg19 to hg38 assembly), outside of DNase clusters +/- 2 kb (ENCFF503GCK, ENCODE database https://www.encodeproject.org/[9]), more than 50 kb away from any transcription start site and outside a gene transcription unit (ENSEMBL Release 103, dataset *hsapiens_gene_ensembl*, http://www.ensembl.org/index.html), more than 300 kb away from cancer-related genes (Cancer Gene Census, GRCh38, COSMIC v92 database https://cancer.sanger.ac.uk/census), more than 300 kb away from any miRNA (ENSEMBL Release 103, dataset *hsapiens_gene_ensembl*, gene_biotype = miRNA, http://www.ensembl.org/index.html) and more than 100 kb away from any long non-coding RNA (ENSEMBL Release 103, dataset *hsapiens_gene_ensembl*, gene_biotype = lncRNA, http://www.ensembl.org/index.html). From the filtered loci we discarded the loci with high BLAT similarity to other sequences. On the RNA-seq level, we required the putative safe harbour sites to be associated with ubiquitously expressed, low variance genes. On the 3D chromosome organization level, they should belong to regions consistently located in active chromosomal compartments across multiple tissue types. Custom code used for in the computational search is available on https://github.com/Foo-Lab.

### Ubiquitously expressed and low-variance genes
We downloaded the median gene-level TPMs by tissue type from GTEx (https://www.gtexportal.org/home/datasets) and identified an empirical set of low-variance housekeeping genes. To this extent, we estimated the mean and the variance of each gene across all available tissue types and, independently, selected the genes with the lowest, insignificant variability using the HVG function of scran R package that decomposes the total variance of each gene into its biological and technical components. We picked the genes whose expression levels do not change significantly across the tissue types (FDR >0.9) and fit a mean vs variance non-parametric lowess regression model. We selected the genes with mean TPM ≥5 and variance below the average (smoothed) variance estimated from the lowess model.

### Loci interaction via chromatin conformation capture
Using a set of publicly available Hi-C chromatin organization data (*Schmitt et al., 2016*) from human cell and tissue types, we shortlisted the genomic regions consistently located (at least 20/21 interrogated tissue types) in active (open chromatin) compartments (*Lieberman-Aiden et al., 2009*).

### BLAT analysis
We measured the uniqueness of target loci using BLAT (https://genome.ucsc.edu/cgi-bin/hgBlat) on the human genome GRCh38 with BLAT'S guess query type. BLAT takes the target DNA sequence as input and identifies similar ones in the whole human genome. Target sequences of more than 25,000

bps (BLAT's limit) were split into multiple smaller overlapping segments of length between 9 000 and 11,000 bps each (depending on the original target length) and tested separately. We calculated the ratio of the BLAT scores for the second hit over the top hit from the BLAT search results. The top hit corresponds to the candidate GSH being tested and the second hit is the most similar sequence match in the genome. We filtered out the candidate loci where the ratio of (BLAT score for 2nd hit) / (BLAT score for 1st hit) was over 0.5 (Supp. File 1).

## Selection of candidate GSH for in vitro validation

Sites for in vitro validation were chosen by running the command *round(runif(1, min = 1, max = 25))* in R to generate seven unique numbers. These numbers were then matched to the ordered list of candidate sites (*Table 1*.) to identify the candidate loci for targeting. Number of sites for targeting was capped at seven due to practical limitations of handling and screening of multiple clonal hES lines simultaneously.

## TAD boundary check

We checked the locations of our in vitro targeted GSH candidates against the TAD borders using data from H1 human embryonic stem cells (hESCs) (*Dixon et al., 2015*) on the 3D Genome Browser (http://3dgenome.fsm.northwestern.edu/; *Wang et al., 2018*). Visual inspection of the candidate loci confirmed that all the candidate GSH are more than 80,000 bp away from TAD borders (*Figure 1—figure supplements 1–7*).

## Plasmid construction

All restriction enzymes were purchased from NEB. PCR reactions were conducted using Q5 Hot Start High-Fidelity 2 X Master Mix (NEB, M0494L). Ligations were conducted using isothermal assembly with NEBuilder HiFi DNA Assembly Master Mix (NEB, E2621L). All gBlocks, primers & oligos were ordered from Integrated DNA Technologies, Singapore. Plasmids used in this manuscript will be made available via Addgene. Primers used for fragment amplification are listed in Supp. File 2.

## pMIA4.721

An in-house expression plasmid containing a CAGG promoter (pMIA4.9) was digested with BamHI & SphI. Two gBlocks (bxb-bsd and bxb-sv) were directly ligated into the digested plasmid. The resulting plasmid was digested with PmlI & KpnI. A fragment containing codon optimised iCasp9-2A-Bsd was amplified from a plasmid supplied by Genewiz (sequence of iCasp9 based on *Straathof et al., 2005*) and ligated into the digested plasmid. This plasmid was subsequently digested with AgeI & SbfI. The SV40 polyA signal was amplified from pMAX-GFP and ligated to the digested plasmid to generate pMIA4.271.

To generate the HDR donors for each GSH candidate, the pMIA4.721 plasmid was digested with NheI for 5' homology arm and with SbfI for 3' homology arm. Homology arms ranging from 240bp to 769bp were amplified from H1 hESC gDNA. Ligation of homology arms was done in two sequential reactions. Order of ligation depended on the underlying sequence of homology arms for each target.

## pMIA22

pMAX-GFP (Lonza) was digested with KpnI and SacI. A gBlock encoding a codon optimised Bxbl-integrase (*Ghosh et al., 2005*) with a C-terminal bi-partite nuclear localisation signal (*Wu et al., 2009*) was amplified and ligated to the digested backbone.

## pMIA10.5-Clover

An empty donor plasmid (pMIA10.5) containing a 5' BxbI attB (CT) and a 3'BxbI attB (GT) was ordered from Genewiz. Clover transgene was amplified and ligated into the plasmid after digestion with AgeI & KpnI.

## pMIA10.53-Clover

We generated a donor plasmid with puromycin selection to allow for enrichment of successfully targeted cells without the need fluorescence activated cell sorting (*Figure 2—figure supplement 9*). pMIA10.5-Clover was digested with BsrGI. A gBlock encoding a stuffer sequence as well as PCR

amplicons of an internal ribosome entry site (IRES) of the encephalomyocarditis virus as well as puromycin N-acetyltransferase were ligated into the digested 10.5-Clover vector to generate 10.53-Clover.

## pMIA10.7

pMIA10.5 empty donor was digested with AgeI & KpnI. Sequences containing the tet-response element and copGFP-SV40pA were amplified from pTRE3G (Clontech, 631173) and pMAX-GFP (Lonza) respectively. The two fragments were ligated to the digested vector to generate an intermediate vector. The intermediate vector was digested with AgeI and an amplicon of Tet-On 3 G (Clontech, 631335) and a gBlock encoding T2A-puromycin-rGlopA were ligated in to generate the pMIA10.7 plasmid.

## Stem cell culture

Human ESC lines H1 (WiCell, WA01) & H9 (WA09, a kind gift from Dr. Lawrence Stanton) were maintained using mTeSR medium (STEMCELL Technologies, 85850) on 1:200 Geltrex (ThermoFisher Scientific, A1413202) coated tissue culture plates and passaged regularly as cell aggregates every 4–5 days using ReleSR (STEMCELL Technologies, 05872). Identity of H1 cells was authenticated by the supplier, WiCell. Identity of H9 cells was authenticated by short tandem repeat analysis (AxilScientific). The cell cultures were tested for mycoplasma contamination monthly and confirmed negative.

## CRISPR/Cas9-mediated targeted construct integration in hESC

H1 hESCs were targeted via nucleofection using an Amaxa-4D (Lonza) as described previously (*Ang et al., 2018*). Briefly, gRNAs were designed using CRISPOR (*Concordet and Haeussler, 2018*; http://crispor.tefor.net/). Three gRNAs with the highest predicted off-target scores and containing a native G-base in the first position were selected for each GSH candidate (Supp. File 2). The gRNAs were cloned into pMIA3 plasmid (Addgene #109399) digested with Esp3I and tested via a GFP reconstitution assay in HEK239T cells. The target locus of each candidate GSH was amplified from H1 hESC gDNA (for primers see Supp. File 2). The amplified target sequences, ranging from 232 to 974 bp were cloned into the pCAG-EGxxFP plasmid (*Mashiko et al., 2013*) (Addgene # 50716, a kind gift from Dr Masahito Ikawa). The pMIA3 with the tested gRNA and the respective pCAG-EGxxFP target plasmid were transfected into HEK293T using lipofectamine3000 (Thermo Fisher Scientific, L3000015), according to manufacturer's recommendations. For each candidate GSH the gRNA with the highest GFP signal at 48 hr post transfection (data not shown) was selected for use in hESC targeting.

Five micrograms of pMIA3 plasmid containing optimal gRNA for each candidate GSH and the respective pMIA4.721 HDR-donor plasmids were nucleofected into hESC using the P3 Primary Cell kit (V4XP-3024) and programme CA-137. A total of $1.5 \times 10^6$ cells were used for each targeting and were plated onto geltrex coated wells on six-well plates in mTeSR with CloneR (STEMCELL Technologies, 05889) following nucleofection. After 24 hr media was changed to mTeSR, and cells were allowed to recover for another 24–48 hr. Once cells reached 70–80% confluency, Blasticidin (ThermoFisher Scientific, A1113903) was added to the culture media at 10 μg/ml. Individual colonies were manually picked from the wells after 7–14 days of selection and expanded further for screening.

## Junction PCR

Genomic DNA samples for all the collected GSH clones was isolated using PureLink Genomic DNA Mini Kit (ThermoFisher Scientific, K182002) according to manufacturer's instructions. PCR reactions amplifying both 5' and 3' targeting HDR junctions as well as the wild type allele were set up using primers listed in Supp. File 2. Samples were checked for the correct amplification size and alignment of the Sanger sequencing reads for each junction PCR and wild type allele.

## Off-target analysis

The top five predicted off-targets were checked via PCR amplification and Sanger sequencing. PCR primers for respective off-targets for each gRNA are listed in Supp. File 2. Sanger sequencing traces covering the off-target site for wild type and the respective targeted clone are shown in *Figure 2—figure supplements 2–4*.

## Copy number analysis

We evaluated Blasticidin and RPP30 Copy Numbers using Droplet Digital Polymerase Chain Reaction (ddPCR) technology (Bio-Rad Technologies), according to manufacturer's specifications. Briefly and following fluorescence-based quantification (Thermo Fisher Scientific, Qubit), 2.5 ng double-stranded DNA was added to a reaction mix containing target-specific primers/probe mixes (900 nM primer/250 nM probe per FAM and HEX fluorophore; Bio-Rad, 10042958 Unique Assay ID: dCNS626289650 and 10031243 Unique Assay ID: dHsaCP2500350), 0.05 U HaeIII Restriction Enzyme (New England Biolabs, R0108S) and ddPCR-specific Supermix for Probes (no dUTP) (Bio-Rad, 1863024). This was randomly partitioned into at least 10,000 discrete oil droplets per reaction using microfluidics within the QX200 Droplet Generator (Bio-Rad, 1864002; together with Droplet Generation Oil for Probes, 1863005), which were gently transferred using a multi-channel pipette into a semi-skirted 96-well plate before heat-sealing (Bio-Rad PX1 PCR Heat Sealer, 1814000). Target amplification within each droplet was conducted in the C1000 Touch Thermal Cycler with 96-Deep Well Reaction Module (Bio-Rad, 1851197) through the following PCR protocol: (1) Enzyme Activation at 95 °C for 10 min, (2) 40 cycles of Denaturation and annealing/extension at 94 °C for 30 s and 55 °C for 1 min, respectively, (3) Enzyme Deactivation at 98 °C for 10 min. The QX200 Droplet Reader (Bio-Rad, 1864003) then derived the number of target-containing droplets through assessing each droplet for elevated, target-specific fluorescence. Blasticidin (FAM)-positive droplet counts were normalised using its respective well-specific RPP30 (HEX)-positive counts prior to downstream analysis. All experiments were done in duplicates, with data visualised and assessed using the QuantaSoft software version 1.7.4.917 (Bio-Rad). We analysed the counts of RPP30 locus, a control locus on chr10 with a copy number of two, and the counts of Blasticidin, positive selection gene included in our landing pad targeting construct (Supp. File 3). We included a no template control, untargeted H1 and H9 cells and previously targeted H1 cells with known single copy integration of Blasticidin as controls for the copy number analysis. Clones whose normalised copy number was approximately 1 were selected for downstream analysis and banking.

## qPCR analysis

RNA was extracted from five biological replicates of the GSH targeted H1 & H9 hESC Pansio-1, Olônne-18 and Keppel-19and two independent cultures of untargeted cells using Direct-zol™ RNA Miniprep kit (Zymo Research, ZYR.R2052). One μg of RNA was converted into cDNA with Superscript IV Vilo MM (ThermoFisher Scientific, 11766050). Quantitative PCR reactions using TaqMan gene expression assays and master mix were used to compare the expression levels of *MAGI3*, *TXNL1*, and *ZNRF4* against reference genes *18* S and *GAPDH* (Thermo Fisher Scientific, Hs00326365_m1; Hs00169455_m1; Hs00741333_s1; HS99999901_S1; Hs03929097_g1 and 4444557). Each reaction was run in three technical replicates on the same reaction plate. Quantitative RT-PCR analysis was done as described previously (*Taylor et al., 2019*) and resulting log2 fold change gene expression data was compared against reference H1 untargeted samples. A one sample Student's t-test was used to determine the statistical significance of whether the mean log2-FC was different than 0 when comparing to untargeted hESC controls.

## RNA-seq library prep

RNA samples described above for qPCR were also used for RNA-seq. RNA concentration and quality were checked with an Agilent 2100 RNA Pico Chip (Agilent, 5067–1513). RNA sequencing libraries were prepared using the TruSeq Stranded Total RNA Sample Prep Kit (Illumina, 20020596) including Ribo-Zero to remove abundant cytoplasmic rRNA. The remaining intact RNA was fragmented, followed by first- and second-strand cDNA synthesis using random hexamer primers. 'End-repaired' fragments were ligated with a unique illumina adapters. All samples were multiplexed and pooled into a single library. Sequencing was done on a HiSeq 4000 to a minimum depth of 50 million 150 bp paired-end reads per biological sample. The raw fastq files are available on ENA under the study accession number PRJEB49564 (https://www.ebi.ac.uk/ena/browser/view/PRJEB49564), accession numbers: ERS16364945-ERS16364998.

## RNA-seq quality control

In all experiments, the raw paired-end reads in fastq format were initially processed with FastQC (https://www.bioinformatics.babraham.ac.uk/projects/fastqc/) for quality control at the base and

sequence level. To remove the PCR duplicates we utilised the FastUniq algorithm (*Xu et al., 2012*). The adaptor trimming was performed by Trimmomatic (version 0.39) (*Bolger et al., 2014*). We quantified the 229,649 annotated human transcripts of GENCODE v35 by Kallisto (version 0.46) (*Bray et al., 2016*) followed by conversion of transcript to raw and TPM-normalized gene counts by the tximport package in R (*Soneson et al., 2015*). In total, 40,198 genes were quantified. Subsequently, we performed QC at the raw gene counts, checking for bad-quality samples having less than 100,000 reads or more than 10% reads mapped to mitochondrial RNA or less than 2000 detected genes. All samples of the various experiments were of high quality and were retained for the main analysis.

## RNA-seq differential expression analysis

The differential expression analysis was conducted by DEseq2 (*Love et al., 2014*) evaluating all pairwise comparisons of the four H1 and H9 test samples (1. Pansio-1 GSH targeted hESC; 2. Olônne-18 GSH targeted hESC; 3. Keppel-19 GSH targeted hESC; 4. Independent untargeted hESC) to an untargeted wild-type H1 or H9 control sample, respectively. In each comparison, we considered only the expressed genes, that is those with non-zero raw counts in at least one sample. The differentially expressed genes were those with $|logFC|\geq 1$ and FDR $\leq 0.01$.

## Functional enrichment analysis

Functional enrichment analysis was performed on the differentially expressed genes using the g:GOSt R package for g:Profiler (version e104_eg51_p15_3922dba) with g:SCS multiple testing correction method applying significance threshold of 0.05 (*Raudvere et al., 2019*).

## Karyotyping

For each H1 and H9 cell line, 20 GTL-banded metaphases were counted, of which a minimum of four have been analysed and karyotyped.

## Teratoma injections

Pansio-1, Olônne-18, and Keppel-19 H1 hESCs from in a 10-cm tissue culture dish (~80% confluency) were dislodged in 3 ml of TeSR-E8 (STEMCELL Technologies, 05990) into small clumps by manual scraping using a serological pipette. Small clumps of hESCs were centrifuged and the pellet was resuspended in 50 µl of TeSR-E8, followed by 50 µl of matrigel (Corning) (after thawing on ice). The hESC suspension was kept on ice until injection. Six to 8 weeks old NOD-SCID mice were used for teratoma studies. Briefly, around 100 µl of hESC suspension was injected intra-muscularly into the gastrocnemius. After about 8 weeks, teratoma was observed and extracted from the mouse. Teratoma samples were fixed in zinc formalin overnight at 4 °C before being sent to the Advanced Molecular and Pathology Laboratory (A*STAR, Singapore) for paraffin embedding and sectioning. Tissue slides were stained with haematoxylin and eosin (H&E) and imaged using the Olympus BX-61 Upright microscope. All animal experiments were reviewed and approved ethics and animal care committees (IRB approval: A*STAR IRB 2020–096 & IACUC: 181366 and 221660).

## Fluorescence activated cell sorting and flow cytometry

H1 and H9 wild type hESCs and respective Pansio-1, Olônne-18, and Keppel-19 lines targeted with pMIA22 and pMIA10.5-Clover were disassociated with accutase (STEMCELL Technologies, 07922) and resuspended in PBS. Cells positive for Clover were enriched with the BD Aria Fusion sorter at the Flow Cytometry Core, SIgN A*STAR. The wild type hESC were used to set up negative gating on the FITC-channel and positive cells were collected in CloneR media supplemented with penicillin-streptomycin (ThermoFisher Scientific, 15140148). After 24 hr of recovery in CloneR the cells were maintained in mTeSR as described above.

For flow cytometry analysis at 5, 10, and 15 passages after Clover integration, H1 and H9 wild type hESCs and Pansio-1, Olônne-18, and Keppel-19 H1 and H9 lines carrying Clover-transgene were disassociated with accutase and resuspended in PBS. The single cells in PBS were analysed with a BD LSR Fortessa x-20 FACS Analyzer and FlowJo (v10.6.1).

## hESC cardiac differentiation

Two days prior to starting differentiation, cells were dissociated using Accutase and seeded as single cells in Geltrex-coated 12-well plates at seeding density between 1 and $1.5\times10^6$ cells. Cardiac

differentiation was performed following the published protocol by *Lian et al., 2013*, with modifications as follows. Six µM of CHIR99021 (STEMCELL Technologies, 72054) was added on day 0 and left for 24 hr followed by medium change. On day 3, 5 µM IWP2 (Sigma-Aldrich, I0536) was added using 50/50 mix of new fresh medium and conditioned medium collected from each well and left for 48 hr. Culture medium from day 0 until day 7 was RPMI1640 (HyClone, SH30027.01) plus B-27 serum-free supplement without insulin (Gibco, A1895601). From day 7 and onwards RPMI1640 with B-27 serum-free supplement with insulin (Gibco, 17504044) was used and changed every 2–3 days.

## hESC differentiation to hepatocyte-like cells

hESCs were differentiated into hepatocyte-like cells as described previously (*Hannan et al., 2013*; *Ng et al., 2019*), with some modifications. Briefly, hESCs were dissociated into small clumps using RelesR and plated onto gelatin-coated coverslips in a 12-well plate with mTeSR. Two days later, hESCs were induced to differentiate into definitive endoderm (DE) cells in RPMI-1640 medium (Gibco) containing 2% B-27 (Invitrogen), 1% non-essential amino acids (Gibco), 1% GlutaMAX (Gibco) and 50 µM 2-mercaptoethanol (Gibco) (basal differentiation medium), supplemented with 100 ng/ml Activin A (R&D Systems), 3 µM CHIR99021 (Tocris) and 10 µM LY294002 (LC Labs) for the first 3 days (D0 to D3). From D3 to D6, cells were incubated in basal differentiation medium supplemented with 50 ng/ml Activin A to form foregut endoderm cells. From D6 to D10, cells were incubated in basal differentiation medium supplemented with 20 ng/ml BMP4 (Miltenyi Biotec) and 10 ng/ml FGF10 (Miltenyi Biotec) to form hepatic endoderm cells. From D10 to D24, hepatic endoderm cells were incubated in HCM Bulletkit (Lonza) differentiation media supplemented with 30 ng/ml Oncostatin M (Miltenyi Biotec) and 50 ng/ml HGF (Miltenyi Biotec). Differentiation medium was replaced every 2 or 3 days.

## hESC neural induction

Human ES cells cultured in mTeSR complete medium for 1–2 days were then used for neural induction as published (*Li et al., 2011*; *Wang et al., 2017*). Briefly, 20–30% confluent hESC were treated with CHIR99021, SB431542 and Compound E in neural induction media, changed every 2 days; 7 days later, the cells were split 1:3 by Accutase and seeded on matrigel-coated plates. ROCK inhibitor (1254, Tocris) was added (final concentration 10 µM) to the suspension at passaging. Cells were then cultured in neural cell culture medium. These derived cells are neural precursor cells (NPC), which were used for further studies.

## Neuronal differentiation

Spontaneous neuronal differentiation was performed as previously described (*Li et al., 2011*). Briefly, the derived $2 \times 10^5$ NPCs were seeded on poly-l-lysine (P4707, Sigma) and laminin (L2020, Sigma) coated six-well plates in neural cell culture medium. The next day, the cells were cultured in neuron differentiation medium: DMEM/F12 (11330–032), Neurobasal (21103–049), 1 X N2 (17502–048), 1 X B27 (17504–044), 300 ng/ml cAMP (A9501), 0.2 mM vitamin C (A4544-25), 10 ng/ml BDNF (450-02), 10 ng/ml GDNF (450-10) until day 30.

## Pancreatic β cell differentiation

Human embryonic stem cells were differentiated to pancreatic β-like cells following a previously published protocol *Pagliuca et al., 2014* with slight modifications. hESCs were dissociated into single cells using TrypLE Express (Gibco, 12605–010) for 3 min and 3–4 million cells were seeded at a density of 1 million cells/mL in mTeSR1 (STEMCELL Technologies, 85850) with 10 µM of Y-27632 (STEMCELL Technologies, 72302) into each well of a non-treated six-well plate. Cells were incubated on a shaker at 80 rpm in a humidified incubator at 37 °C with 5% $CO_2$. The next day, the differentiation was initiated and carried out in five stages (S1, S2, S3, S5, and S6) with the media composition listed in *Supplementary file 2*. Media were changed every other day if the same type of media was required.

## Flow cytometry analysis for pancreatic cells

At D35 of β cell differentiation, hESC-derived β-like cells were dissociated using TrypLE Express for 5 min. The TrypLE Express was then diluted 4 x with 10% FBS in PBS. Single cells were enriched by passing cell suspension through a 40 µm filter. After washing once with PBS, the cells were fixed in 4% PFA for 15 min before blocking in 5% FBS in PBS with 0.1% Triton X-100 (Merck Millipore, 9410).

Cells were stained with primary antibody (Supp. File 2) for 1 hr at room temperature, followed by secondary antibody (Supp. File 2) for 1 hr at room temperature. Flow cytometry was performed using the BD FACSymphony analyser. 10,000 events were collected for each sample. FlowJo v10 software was used for analysis.

## Immunofluorescence

Cells on coverslips were fixed in 4% paraformaldehyde (Wako) for 15 min at room temperature, before blocking in 5% donkey serum (EMD Millipore) in PBS with 0.1% Triton X-100 for 1 hr at room temperature. Cells were stained with primary antibody overnight at 4 °C (see Supp. File 2 for antibodies used), or for control slides with blocking buffer. Secondary antibody staining was done with the appropriate AlexaFluor 594 for 1 hr at room temperature. Lastly, cells were stained with DAPI (Sigma-Aldrich, 1:5000) for 20 min at room temperature. Coverslips were mounted onto glass slides using Vectashield (Vector Laboratories). Images were taken using the EVOS M5000 microscope. Light intensity and gain were kept consistent across samples and controls with each antibody.

## High-content imaging (HCI)

For high content imaging of Pansio-1, Olônne-18, and Keppel-19 H9 clones were differentiated into neuronal, hepatocyte and cardiac cell types as described above, with minor adjustments. Differentiation to neuronal and hepatic cell types was conducted directly on the HCI 96-well plates (Cell-Carrier Ultra, PerkingElmer) by seeding undifferentiated cells into the plates and proceeding with differentiation protocol scaled to 96-wells. Differentiation to cardiac cells was conducted as above. Cells at day 14 of the protocol were disassociated and seeded onto the 96-well imaging plates as follows. Cells were incubated with 1 mg/ml Collagenase IV (Worthington Biochemical Corporation, LS004186) for approximately 30 min followed by an approximately 15 min incubation with 1:1 mixture of accutase and trypsin (ThermoFisher, 15400054). Once the cells were disassociated, they were spun down and supernatant was discarded. The cells were resuspended in 80% RPMI1640 with B-27 serum free supplement with insulin and 20% foetal bovine serum (ThermoFisher, 16000044) supplemented with ROCK inhibitor (STEMCELL TECHNOLOGIES, Y-27632) at 5 µM and seeded onto the imaging plates coated with geltrex. 24 hr after seeding the media was changed to normal cardiac media. Each of the clones was plated in six replicate wells for each different lineage.

Blocking and staining for lineage markers was done as described above with minor adjustments. The steps were conducted on the imaging plates in 100 µl volumes. Three wells of each clone were incubated with antibodies against the respective lineage markers and three wells were incubated with an isotype control. After secondary antibody staining the cells were incubated with Hoechst 33342 (Thermo Fisher, H3570) at 1:2500 dilution for 15 min. The stained plates were kept at 4 °C in 0.2% PFA in PBS and shielded from light until imaging.

HCI was done using a PerkinElmer Opera Phenix confocal imager with a 20 x objective. Sixty-one fields were imaged for each well and the channels Alexa 488, Alexa 594 and HOECHST 33342 were recorded. Image analysis was conducted on PerkinElmer Columbus software. Details of the analysis pipelines used are found in the appendix. The raw HCI images are available on Dryad (https://doi.org/10.5061/dryad.p8cz8w9ww).

## Generation and targeting of inducible GSH cells

H1 Pansio-1 cells were targeted with pMIA22 and pMIA10.7 constructs. After 48 hr of recovery puromycin (ThermoFisher, A1113803) was added at 1 µg/ml. The cells were selected for approximately 72 hr to ensure removal of untargeted cells. For induction of transgene expression, the cells were cultured with 1 mg/ml doxycycline (MerckMillipore, D5207-1G) for 24 hr before analysis.

The newly generated H1 and H9 safe harbour cell lines Pansio-1, Olônne-18, and Keppel-19 are available from the corresponding authors upon request and under an MTA and/or RCA, and will be made available through the WiCell repository. The plasmids used in this study will be made available through Addgene.

## Acknowledgements

This manuscript is dedicated to the memory of Seppo J Autio. We would like to express our gratitude to Dr Shyam Prabhakar, Dr Alexander Lezhava and Dr Jay Shin for their generous support. We thank

the late Albert Barillé for life-long inspiration. We would also like to thank Dr Steve Oh and Dr Ludovic Vallier for insightful discussions and feedback on the manuscript, as well as Dr Lee Siggens for feedback and assistance on primer design. The authors thank the Agency for Science, Technology and Research (A*STAR)'s Singapore Immunology Network (SIgN) Flow Cytometry platform for enabling the flow cytometry work for this publication. A*STAR's SIgN Flow Cytometry platform is supported by SIgN, A*STAR, and the National Research Foundation (NRF), Immunomonitoring Service Platform (Ref: ISP: NRF2017_SISFP09) grant. The authors would like to acknowledge the invaluable support provided by the EDDC Academic Research Organisation (EARO) High Throughput Phenomics Platform and their staff for enabling the high content imaging experiments in this publication. This research was funded by the BMRC YIG 2016 (1610851033) and A*STAR CDF 2020 (202D8020) to M.I.A. The funding bodies played no role in the design of the study or in collection, analysis, and interpretation of data or in writing of the manuscript.

## Additional information

### Competing interests

Matias I Autio, Efthymios Motakis, Arnaud Perrin, Roger SY Foo: Patent application PCT/SG2022/050888. The other authors declare that no competing interests exist.

### Funding

| Funder | Grant reference number | Author |
|---|---|---|
| Biomedical Research Council | 1610851033 | Matias I Autio |
| Agency for Science, Technology and Research | 202D8020 | Matias I Autio |

The funders had no role in study design, data collection and interpretation, or the decision to submit the work for publication.

### Author contributions

Matias I Autio, Conceptualization, Data curation, Software, Formal analysis, Supervision, Funding acquisition, Investigation, Visualization, Methodology, Writing – original draft, Project administration, Writing – review and editing; Efthymios Motakis, Software, Formal analysis, Writing – review and editing; Arnaud Perrin, Zenia Tiang, Dang Vinh Do, Joanna Tan, Shirley Suet Lee Ding, Wei Xuan Tan, Investigation, Writing – review and editing; Talal Bin Amin, Software, Writing – review and editing; Jiaxu Wang, Resources, Investigation, Writing – review and editing; Chang Jie Mick Lee, Investigation; Adrian Kee Keong Teo, Resources, Supervision, Writing – review and editing; Roger SY Foo, Formal analysis, Supervision, Writing – original draft, Writing – review and editing

### Author ORCIDs

Matias I Autio (ID) https://orcid.org/0000-0001-9579-9617
Arnaud Perrin (ID) https://orcid.org/0000-0003-3545-5470
Adrian Kee Keong Teo (ID) https://orcid.org/0000-0001-5901-7075

### Ethics

All animal experiments were reviewed and approved ethics and animal care committees (IRB approval: A*STAR IRB 2020-096 & IACUC: 181366 and 221660).

### Decision letter and Author response

Decision letter https://doi.org/10.7554/eLife.79592.sa1
Author response https://doi.org/10.7554/eLife.79592.sa2

## Additional files

### Supplementary files

• Supplementary file 1. Output of safe and active searches and their overlap. (1) Coordinates that pass safe site filters. (2) Regions that pass active filters. (3) Overlap of safe sites and active regions.

• Supplementary file 2. Oligos, primers and antibodies used, and details of beta cell differentiation. (1) gRNAs used for targeting. (2) Primers used for plasmid construction. (3) Primers used for junction PCR and WT allele screening. (4) Primers used for predicted off-target screening. (5) Antibodies used for immunofluorescence staining and flow cytometry. (6) Details for pancreatic differentiation.

• Supplementary file 3. GSH targeted hESC screening. (1) Summary of GSH targeted hESC clones screening. (2) ddPCR copy number analysis.

• Supplementary file 4. Source data for qPCR results. (1) qPCR data source file for *Figure 2C*. (2) qPCR data source file for *Figure 2—figure supplement 5A*.

• Supplementary file 5. RNAseq analysis of GSH targeted H1 clones. (1) Differentially expressed genes |logFC|≥1 and FDR ≤0.01. (2) Functional enrichment analysis results from g:GOSt.

• Supplementary file 6. RNAseq analysis of GSH targeted H9 clones. (1) Differentially expressed genes |logFC|≥1 and FDR ≤0.01. (2) Functional enrichment analysis results from g:GOSt.

• Supplementary file 7. HCI data analysis source file for *Figure 4—figure supplement 1*.

• Supplementary file 8. Percentage of FITC-A positive cells from H1 Pansio inducible cells source file for *Figure 2—figure supplement 8*.

• MDAR checklist

### Data availability

Unprocessed RNAseq FASTQ files generated for this study are available from ENA: PRJEB49564 accession numbers: ERS16364945-ERS16364998. Custom computational scripts used for the GSH search is available from https://github.com/Foo-Lab/safeharbour-sites (copy archived at *Foo-Lab, 2024*). High content imaging data available is on Dryad at https://doi.org/10.5061/dryad.p8cz8w9ww. All other data generated during this study are included in the manuscript and *Supplementary files 1–8*.

The following datasets were generated:

| Author(s) | Year | Dataset title | Dataset URL | Database and Identifier |
|---|---|---|---|---|
| Autio MI | 2021 | putative genomic safe harbour loci for transgene expression in human cells | https://www.ebi.ac.uk/ena/browser/view/PRJEB49564 | ENA, PRJEB49564 |
| Autio MI | 2023 | Computationally defined and in vitro validated putative genomic safe harbour loci for transgene expression in human cells | https://doi.org/10.5061/dryad.p8cz8w9ww | Dryad Digital Repository, 10.5061/dryad.p8cz8w9ww |

The following previously published dataset was used:

| Author(s) | Year | Dataset title | Dataset URL | Database and Identifier |
|---|---|---|---|---|
| The Genotype-Tissue Expression (GTEx) Project | 2017 | Median gene-level TPM by tissue. Median expression was calculated from the file | https://www.gtexportal.org/home/downloads/adult-gtex/bulk_tissue_expression | GTEx Portal, GTEx_Analysis_2017-06-05_v8_RNASeQCv1.1.9_gene_tpm.gct.gz |

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
