## [Editor Report]

This study presents solid data on the computational identification of 25 putative human genomic safe harbor loci, of which 3 have been experimentally validated using human embryonic stem cells, that may serve as safe sites for transgene integration. These findings will be an invaluable resource in cell and gene therapy applications. This work will be of interest to cell biologists and researchers in the stem cell field.

---

## [Decision Letter]

[Editors' note: this paper was reviewed by Review Commons.]

---

## [Author Response]

General Statements

We thank all the reviewers for their comments and suggestions, which we have addressed below. In addition to the revisions relating to the reviewer comments we have also provided further experimental support for our manuscript.

Some of the reviewer suggestions include extensive engineering of new hES lines for further validation and comparison of our candidate GSH. We value these suggestions and have made plans to complete some of them, however a subset of the experiments suggested are not possible for us to complete due to time and man power constraints. We have detailed our reasoning for the experiments that can and cannot be accomplished below. The engineering of new hES lines does take considerable time and effort, hence we estimate that the experiments that we have planned will take us approximately 6-months to complete.

Description of the planned revisionsReviewer #12. please add data that quantifies the number of cells expressing Clover, in the differentiated cell types. Ideally, multiple markers for each lineage should be used.

We thank the reviewer for this suggestion and agree that quantification of Clover positive cells in the lineage differentiation is a valuable addition to our manuscript. We will repeat our differentiation protocols and will include quantification of Clover expression using high content imaging and/or flow cytometry. We will also include more lineage marker proteins to further validate the differentiated phenotypes of our cells.

Reviewer #21) They only examined expression from the CAG promoter unit. However, this does not guarantee stable expressions from other promoters. Since the CAG promoter is very strong, it may be resistant against cellular silencing activity. For research purpose, tetracycline-regulatable promoters are often used, and it has been reported that although CAG promoter is not silenced, the TRE promoter is silenced when an expression unit is placed at AAVS1 locus (Ordovas L et al. Stem Cell Rep, 5: 918-931, 2015). Therefore, before concluding that these loci are GSH, expression from the TRE promoter should be tested.

We appreciate the reviewer raising this point. Indeed testing a promoter different than CAG in our candidate GSH is an important experiment. We will use our established targeting protocols to engineer a TRE driven expression cassette into one of our candidate GSH and will examine the activity in hES and derived cell types.

Additional experiments (not suggested by reviewers)

To provide additional resources for the research community and to account for variability seen between different pluripotent stem cell lines we have also targeted and established landing pad expression lines in H9 hESC cells for Pansio-1, Olônne-18 and Keppel-19 GSH. We are currently conducting similar validation experiments for these lines as presented for the H1 cells in the current version of the manuscript. The data from the H9 cells will be incorporated in the revised version of the manuscript.

Description of the revisions that have already been incorporated in the transferred manuscriptReviewer #11. Please expand on the rationale for selection of the seven sites that were selected for initial targeting (i.e. what differentiated these from the other 18 sites as being suitable), and on the results for why no successful edits were identified for 4 of these loci.Reviewer #3The second major issue is that it is unclear how the authors picked seven loci for more extensive targeting out of the 25 initially identified. Without knowing the criteria used for these selections, it is challenging to know if there was a bias in selection of sites for further analysis that could alter results, or if the other identified sites are truly acceptable targets. In addition, only three of those GSH sites were successfully targeted. As a result, it is hard to determine the validity of the authors claim that they identified 25 unique GSH loci when they only fully characterize three of them. While it is not necessary to test all 25, it might be beneficial to test more than three before making these conclusions.

We apologise for not making the selection criteria for our in vitro experiments clearer in the manuscript. To clarify, the 7 sites were chosen at random from the 25 candidate sites (by running a command in R *round(runif(1, min = 1, max = 25))*.

We capped the selection at 7 candidate loci due to the considerable amount of hands-on cell culture work in growing, CRISPR-targeting and screening clones in hES cells. We established successfully targeted clones from 3 candidate GSH in the single round of targetings done. It is likely that with further rounds of targeting and possible optimisation we would be able to generate positive clones from the remaining candidate GSH, however as a result of practical limitations of time and man power we proceeded with the characterisation of the three established sites. While it is true that we do not have validation data of all 25 candidate sites, all the sites are the result of the same bioinformatic selection and hence we class them as *candidate GSH*. Between the detailed curation of possible safe harbour sites and the generation of usable and functional safe harbour stem cell lines, there are a number of practical issues such as the efficiency of CRISPR targeting, which fall outside the present studies scope. Therefore after successful targeting of 3 candidates we proceeded with characterisation of the successfully engineered sites.

We have edited the main text and the methods section to clarify the selection of target sites.

Main text page 4

“To validate our candidate GSH, we selected seven of the 25 sites at random for in vitro experiments (Figure 1 C).”

Supp methods page 2

“Selection of candidate GSH for in vitro validation

Sites for in vitro validation were chosen by running the command round(runif(1, min = 1, max = 25)) in R to generate seven unique numbers. These numbers were then matched to the ordered list of candidate sites (Table 1.) to identify the candidate loci for targeting. Number of sites for targeting was capped at seven due to practical limitations of handling and screening of multiple clonal hES lines simultaneously.”

Reviewer #22) They examined off-target integration by PCR and Sanger sequencing of the top 5 predicted off target sites. However, Southern blot analyses are needed to rule out off-target integrations. (This reviewer cannot evaluate data of copy number analysis using Digital PCR).

We agree with the reviewer that Southern blotting is a tried and tested method for evaluating integration of transgenes. However, Southern blotting is also a very labour intensive method and requires the use of radio labelled probes. Digital PCR has been extensively verified as a reliable method to accurately evaluate copy number of transgenes and native gene loci (Bell *et al.* 2018, *Meth in Mol Biol*; Collier *et al.* 2017, *Plant J*; Hu *et al.* 2020, *Jour Trans Med*, Härmälä *et al.* 2017, *Meth in Mol Biol*). We used droplet digital PCR to evaluate the copy number of our landing pad construct (assay targeting the Blasticidin gene) and discarded any clones which had multiple copies. Using PCR and Sanger sequencing of the target locus we established that all the clones had only a single copy integration at the target site, given that all clones amplified a wild-type allele with primers flanking the gRNA target (Supp Figure 2). Hence any clone with a copy number of more than ~1 was discarded as they contained an off-target integration (Supp Table 2).

PCR and Sanger sequencing of the top 5 predicted off-targets was conducted to check for off-target cutting by CRISPR/Cas9, which was not evident at any of the analysed sites.

We have now edited the methods section and supplementary table 3 to clarify the process of clone screening with ddPCR

Supp methods page 4

“We analysed the counts of RPP30 locus, a control locus on chr10 with a copy number of two, and the counts of Blasticidin, positive selection gene included in our landing pad targeting construct (Supp. Table 3). We included a no template control, untargeted H1 cells and previously targeted H1 cells with known single copy integration of Blasticidin as controls for the copy number analysis. Clones whose normalised copy number was approximately 1 were selected for downstream analysis and banking.”

Reviewer #3First, it is unclear how the authors used BLAT to narrow down their initial list of 49 safe loci down to 25. A more detailed explanation in the text (vs methods) would aid in reader understanding of methodology.

We are sorry that the details of our filtering with BLAT were not clear in the original version of the manuscript. We have now edited the main text, the methods section and Supp Table 1. to clarify the process.

Main text page 3

“We further filtered the 49 sites (Figure 1 A-B, Supp. Table 1) using BLAT to remove candidate sites that had highly similar sequence matches at other genomic loci and generated a final shortlist of 25 unique putative universal GSH sites in the human genome (Figure 1 A-B, Table 1).”

Supp methods page 2

“We calculated the ratio of the BLAT scores for the second hit over the top hit from the BLAT search results. The top hit corresponds to the candidate GSH being tested and the second hit is the most similar sequence match in the genome. We filtered out the candidate loci where the ratio of (BLAT score for 2nd hit) / (BLAT score for 1st hit) was over 0.5 (Supp. Table 1).”

In addition, a deeper explanation of how differentially expressed genes were identified would be helpful. The authors state that many of the DE genes in their GSH targeted loci were identical to those found in both control and untargeted cells. It is unclear what the comparator was in these experiments that was used to identify those DE genes; clarification of this in the text would be helpful for the reader.

We thank the reviewer for raising this point and apologise for not including enough details on the DE gene analysis in our original manuscript. Briefly, the analysis was conducted by pairwise comparisons of three biological replicates from 5 test cell lines to an untargeted H1 WT control cell line. The 5 test samples are as follows:

**Author response table 1. sa2table1:** 

Olônne-18 GSH targeted H1 hESC
Keppel-19 GSH targeted H1 hESC
Control targeted H1 hESC (CRISPR and HDR of an expression cassette at a non-GSH locus)
Independent untargeted H1 hESC

In each comparison, we considered only the expressed genes, i.e. those with non-zero raw counts in at least one sample. The differentially expressed genes were those with |logFC| ≥ 1 and FDR ≤ 0.01. The lists of DE genes from each comparison are presented in Supp. Table 4. Overlap between the DE genes in GSH comparisons and with the Control or WT H1 comparisons are highlighted by the yellow, orange and blue highlights. A summary of these overlaps is presented in Figure 2E.

We have edited both the main text and the methods section to provide more clarity on the DE analysis

Main text page 6

“We then conducted RNA-seq analysis to look for gene expression changes on a global scale. Pairwise comparisons of our GSH targeted clones to un-targeted H1 hESCs yielded very low numbers of differentially expressed (DE) genes; 30, 29 and 40 respectively for Pansio-1, Olônne-18, and Keppel-19 (Figure 2 D and Supp. Table 4). Notably CASP9, the suicide gene included in our targeting construct, was the gene with lowest false discovery rate (FDR) in Pansio-1 and Olônne-18 and second lowest in Keppel-19 (Figure 2 D and Supp. Table 4). A high proportion of the DE genes found in our GSH targeted lines were shared with the DE genes found from pair-wise comparisons of control targeted H1 cells and an independent set of wild-type H1 cells to the un-targeted H1 hESCs (Figure 2 E and Supp. Table 4) Suggesting that the observed changes were unrelated to the GSH targeting and transgene expression.”

Supp methods page 5

“The differential expression analysis was conducted by DEseq2 (Love et al., 2014) evaluating all pairwise comparisons of the five test samples (1. Pansio-1 GSH targeted H1 hESC; 2. Olônne-18 GSH targeted H1 hESC; 3. Keppel-19 GSH targeted H1 hESC; 4. Control targeted H1 hESC 5. Independent untargeted H1 hESC) to an untargeted wild-type H1 control sample.”

In figure 2, the labeling of the panels is quite confusing, as panel F appears between panels E and D.

Additional data & experiments (not suggested by reviewers)

To further validate that targeting our GSH sites does not interfere with pluripotency of the hES cells we conducted teratoma assays with our three GSH targeted H1 hES lines. All three GSH lines were able to give rise to cells from endoderm, ectoderm and mesoderm lineages corroborating results from our directed differentiation experiments. We have included the data from the teratoma experiments in Supp Figure 4 and added a section both to the main text as well as methods.

Main text page 6

“We also used the Pansio-1, Olônne-18 and Keppel-19 H1 hES lines to generate teratomas in immunodeficient mice. All three lines were able to give rise to tissues from endoderm, ectoderm and mesoderm lineages (Supp. Figure 4). Taken together, these results suggest minimal disruption of the native genome or the pluripotent cell identity and show a lack of karyotypic abnormalities following transgene integration to our three GSH.”

Supp methods page 5 & 6

“Teratoma injections

Pansio-1, Olônne-18, and Keppel-19 H1 hESCs from in a 10 cm tissue culture dish (~80% confluency) were dislodged in 3ml of TeSR-E8 (STEMCELL Technologies, 05990) into small clumps by manual scraping using a serological pipette. Small clumps of hESCs were centrifuged and the pellet was resuspended in 50ul of TeSR-E8, followed by 50ul of matrigel (Corning) (after thawing on ice). The hESC suspension was kept on ice until injection. 6-8 weeks old NOD-SCID mice were used for teratoma studies. Briefly, around 100ul of hESC suspension was injected intra-muscularly into the gastrocnemius. After about 8 weeks, teratoma was observed and extracted from the mouse. Teratoma samples were fixed in zinc formalin overnight at 4°C before being sent to the Advanced Molecular and Pathology Laboratory (A*STAR, Singapore) for paraffin embedding and sectioning. Tissue slides were stained with haematoxylin and eosin (H&E) and imaged using the Olympus BX-61 Upright microscope. All animal experiments were reviewed and approved ethics and animal care committees (IRB approval: A*STAR IRB 2020-096 & IACUC: 181366 and 221660).”

We also added edited the methods section to add details of fluorescence activated cell sorting, which was used to enrich the positive cells at passage 1 after BxbI mediated Clover targeting. Flow analysis was conducted on these cells 5, 10 and 15 passages after the sorting.

Supp methods page 6

“Fluorescence activated cell sorting and flow cytometry

H1 wild type hESCs and Pansio-1, Olônne-18, and Keppel-19 H1 lines targeted with pMIA22 and pMIA10.5-Clover were disassociated with accutase (STEMCELL Technologies, 07922) and resuspended in PBS. Cells positive for Clover were enriched with the BD Aria Fusion sorter at the Flow Cytometry Core, SIgN A*STAR. The H1 wild type hESC were used to set up negative gating on the FITC-channel and positive cells were collected in CloneR media supplemented with penicillin-streptomycin (ThermoFisher Scientific, 15140148). After 24h of recovery in CloneR the cells were maintained in mTeSR as described above.

For flow cytometry analysis at 5, 10 & 15 passages after Clover integration, H1 wild type hESCs and Pansio-1, Olônne-18, and Keppel-19 H1 lines carrying Clover-transgene were disassociated with accutase and resuspended in PBS. The single cells in PBS were analysed with a BD LSR Fortessa x-20 FACS Analyzer and FlowJo (v10.6.1).”

Description of analyses that authors prefer not to carry outReviewer #13. Functional studies of the differentiated cell types would add substantial value to this paper. in the absence of such data, additional marker proteins that reflect functional properties or the maturity of the derived cell types could be added.

While we agree with the reviewer that functional studies using the GSH are of interest, setting up such assays in multiple different cell types is beyond the time and resources available for the revisions. Furthermore, we feel that such experiments are beyond the focus of the current manuscript. We will be including additional marker proteins for each of the differentiation phenotypes assayed (as detailed above), and will also include recordings of functional beating cardiomyocytes in the revised manuscript.

Reviewer #3One major question is whether or not these safe genomic loci identified are actually better than traditionally used loci such as Rosa26 or CCR5. While the authors note in the Discussion section that these traditional loci do not meet their criteria for a safe genomic integration site, they do no direct comparison of their new loci vs these more traditional ones. A side-by-side comparison would make the data more convincing that these loci are better suited for genomic integration (such as noting fewer changes in the transcriptome etc).

As the reviewer points out the traditionally used sites targeting sites are Rosa26 and CCR5 (and also AAVS1). These sites have been used widely in the research setting, however as discussed by Sadelain *et al.* 2012 *Nat. Rev. Cancer* and Papapetrou & Schambach 2016 *Mol. Ther.* each of the three traditional sites has a number of drawbacks (including disruption of coding or non-coding genes and proximity to oncogenes). The points highlighted in the above papers may limit application of the traditional sites in a clinical setting. The criteria used to define the GSH in this study specifically avoid gene transcription units as well as setting a limit on the distance from known oncogenes, as such our candidate sites by definition will not have the potential issues highlighted for the traditional sites.

While we agree that a direct comparisons between our new GSH and the traditional sites are of interest, these comparisons would require a very significant amount of work. We would have to establish targeting at the traditional sites with our own donor constructs and then target hESC, screen for correct clones, establish the cell lines, followed by all the analyses (e.g. RNAseq, transgene expression, lineage differentiations). This would likely require close to or possibly even more than one year to accomplish. Unfortunately this is beyond what is possible with the man power, resources and time available for the revision.

In figure 2, the labeling of the panels is quite confusing, as panel F appears between panels E and D.

We apologise for the confusion caused by the figure labelling. This is simply due to sizing constraints of fitting the figure on a single page. If preferred this can be amended in a revised version of the manuscript.

Finally, in figure 3, while the three new cell lines are shown to be differentiated into various cell types, no control images are shown for comparison. This would be helpful to add in

The lineage differentiation protocols (Li et al., 2011 *PNAS*; Wang et al., 2017 *Gen. Res.*; Lian et al., 2013 *Nat. Prot*.; Hannan et al., 2013 *Nat. Prot.*; Ng et al., 2019 *iScience*) used in our manuscript are well established, widely utilised and shown to work with WT hES and iPS cells. Our differentiation experiments were conducted with the aim of determining whether we can generate cells from the three germ lineages following targeting of a candidate GSH and we establish that with the positive staining for lineage markers. As it is already known that the wild type H1 cells can give rise to these cell types we feel that it is unnecessary to include these.